# Cd, Cu, and Zn Accumulations Caused by Long-Term Fertilization in Greenhouse Soils and Their Potential Risk Assessment

**DOI:** 10.3390/ijerph16152805

**Published:** 2019-08-06

**Authors:** Zhongbin Liao, Yali Chen, Jie Ma, Md. Shafiqul Islam, Liping Weng, Yongtao Li

**Affiliations:** 1Agro-Environmental Protection Institute/Key Laboratory for Environmental Factors Control of Agro-product Quality Safety, Ministry of Agriculture and Rural Affairs, Tianjin 300191, China; 2College of Natural Resources and Environment, South China Agricultural University, Guangzhou 510642, China

**Keywords:** heavy metal, greenhouse soils, chemical fraction, geo-accumulation index, principle component analysis

## Abstract

The intense management practices in greenhouse production may lead to heavy metal (HM) accumulations in soils. To determine the accumulation characteristics of HM and to evaluate possible HM sources in greenhouse soils, thirty typical greenhouse soil samples were collected in Shouguang District, Shandong Province, China. The results indicate that the Cd, Cu, and Zn concentrations are, respectively, 164.8%, 78.6%, and 123.9% higher than their background values. In the study area, Cd exhibits certain characteristics, such as wide variations in the proportion of its exchangeable form and the highest mobility factor and geo-accumulation index, which are indicative of its high bioavailability and environmental risk. In addition, there is a significant positive correlation between pairs of Cd, P, soil organic carbon, and cultivation age. Combined with principal component analysis, the results indicate the clear effects that agricultural activities have on Cd, Cu, and Zn accumulation. However, Cr, Ni, and Pb have a significant correlation with soil Fe and Al (hydr)-oxides, which indicates that these metals mainly originate from parent materials. This research indicated that long-term intensive fertilization (especially the application of chemical fertilizers and livestock manure) leads to Cd, Cu, and Zn accumulation in greenhouse soils in Shouguang. And the time required to reach the maximum permeable limit in agricultural soils for Cd, Cu, and Zn is 23, 51, and 42 years, respectively, based on their current increasing rates.

## 1. Introduction

For the last two decades in China, greenhouse vegetable cultivation has increased from 1.395 million hm^2^ to over 4.1 million hm^2^ [1,2]. Greenhouse cultivation has contributed significantly to vegetable production. However, this intensive agricultural practice is the cause of several environmental problems due to its high multiple cropping index and productivity. Agricultural activies (e.g., through excessive use of fertilizers, manures, agrochemicals, and irrigation) are possibly the main contributor to heavy metal (HM) accumulation in greenhouse soils [3,4], which ultimately affects human health via food production [5,6].

The Shouguang District is the largest base for greenhouse vegetable production in China, with an annual vegetable production of 4 × 10^6^ tons [7]. Intensive management practices (e.g., more fertilizers and agrochemicals application, high multiple cropping index) in the greenhouse industry in this region have lasted for more than three decades, which has affected HM spatial distributions [8] and led to excessive accumulation of HMs [9] in greenhouse soils. Higher HM accumulations in greenhouse soils than in arable soils were also reported [4,10]. Among the various risk assessment methods used to evaluate potential environmental risks of HMs accumulated in soils, geo-accumulation index (*I_geo_*) is widely used, which indicated higher environmental risks of Cd, Cu and Zn than other HMs in greenhouse soils [11,12,13].

However, *I_geo_* or other assessment methods, such as the ecological risk index and Nemerow multi-factor pollution index, are insufficient indicators of HM bioavailability, mobility, or toxicity. Meanwhile, the majority of previous studies on greenhouse soils in Shouguang or other regions have mainly focused on the total HM content with little emphasis on their chemical speciation. In general, actual environmental risks of HMs depend strongly on the chemical speciation of the HMs in the ambient environment [14,15,16,17]. According to the sequential chemical extraction method proposed by Tessier, et al. [18], the exchangeable fraction represents weakly bound metals and considered as the fraction that is most mobile and readily absorbed by plants. The carbonate-bound fraction is susceptible to changes in pH, whereas the Fe-Mn oxide-bound and organically bound fractions are unstable when exposed to strong reductive and oxidation conditions, respectively. The residual fraction is insoluble under natural conditions, with extremely low bioavailability. Thus, the sum of the exchangeable, carbonate-bound, Fe-Mn oxide-bound, and organically bound fractions is used to characterize the potential bioavailability of a HM, and the sum of the exchangeable and carbonate-bound fractions is known as the mobility factor (MF) [19,20,21]. Thus, the chemical fractions of HMs, which correspond to their chemical binding forms, can provide more useful information when attempting to assess the environmental risks of HMs.

In general, soil HMs derive from parent materials and can be further enriched by human activities [22]. The main anthropogenic sources of HMs in greenhouse soils include chemical fertilizers, manure, sewage sludge, agricultural chemicals and irrigation water [4]. In addition, how to take appropriate actions to reduce HM inputs to soils requires information on the importance and extent of HM contaminations from above-mentioned sources. The most common method to identify main HM sources is to use multivariate statistical analysis, such as correlation analysis and principal component analysis (PCA), to classify metals and selected edaphic parameters (e.g., Al_2_O_3_ and Fe_2_O_3_) into groups and perform a simple comparison with background values (BVs) of metals [6,23,24,25,26,27].

Therefore, in this study, in order to determine HM (including Cd, Cr, Cu, Ni, Pb, and Zn) pollution characteristics and accurately assess their potential risks and clarify their possible sources in Shouguang greenhouse soils, a field survey was conducted with following objectives: (1) analyze the HM concentrations and fractions in greenhouse soils, (2) assess HM accumulations and potential environmental risks, and (3) identify HM sources for targeted reduction of HM inputs to greenhouse soils in Shouguang.

## 2. Materials and Methods

### 2.1. Study Area Description

Shouguang District (36°41′–37°19′ N, 118°32′–119°10′ E) is located in the northern part of the Shandong Province, which is in the lower reaches of the Xiaoqing River and along the southwest coast of Laizhou Bay, Bohai Sea. Shouguang District comprises a total land area of 2.28 × 10^3^ ha, which accounts for 1.43% of the total area of the Shandong Province. Shouguang belongs to the warm temperate monsoon area and has a continental climate, which is characterized by a mean annual precipitation of 608 mm, a frost-free period of 179 to 227 days, and a mean annual temperature of 12.4 °C. The parent soil materials are mainly alluvial and marine sediments, with lake sediments in certain areas, and the soil type is cinnamon soil. Shouguang has unique conditions for vegetable production due to its flat terrain, fertile soil, abundant sunlight, proper temperature, abundant underground freshwater resources, and suitable climate. In the study area, cultivation ages of greenhouses vary from 1 to 30 years. Meanwhile, N-P-K compound fertilizers and livestock manures were frequently applied (Appendix A), leading to excessive fertilizer application and unbalanced nutrient inputs in greenhouse soils [28].

### 2.2. Sampling and Pretreatment

Thirty sites in the southwestern region of Shouguang were randomly selected for greenhouse surface soil (0–20 cm) sampling, mainly distributed in the greenhouse cultivation area of Shouguang (Figure 1). Based on The Technical Specification for Soil Environmental Monitoring (HJ/T 166-2004) [29], large debris, stones, and plastic films were removed from all samples after they were air-dried at room temperature. Samples were then sieved through a 2-mm polyethylene sieve and portions of the sample were ground and sieved through a 0.075-mm polyethylene sieve for further analysis.

### 2.3. Soil Analysis

Soil pH was measured in a 1:5 (w/v) ratio of soil to 0.01 mol·L^−1^ CaCl_2_. For soil organic carbon (SOC) measurement, soil samples were first treated with 2 mol·L^−1^ HCl to remove inorganic carbon, and then measured by total organic carbon analyzer (muti N/C 3100, Analytik Jena, Jena, Germany).

The total contents of HMs, that is Cd, Cu, Zn, Pb, Ni, and Cr, as well as major elements like Fe, Al, K, P in the soils were determined by digesting the soil samples according to the method recommended by USEPA [30] in a 1:10 (w/v) ratio of soil to HNO_3_. A soil reference material (GBW07402) was used for quality control during analysis.

The sequential chemical extraction was performed based on the method proposed by Tessier et al. [18], which defined HM fractions as water-soluble/exchangeable form (F1), carbonate-bound form (F2), Fe-Mn oxide-bound form (F3), organically bound form (F4), and residual form (F5).

Cu, Zn, Pb, Ni, Cr, Fe, Al, K, and P concentrations in all extracts (including digestion and sequential chemical extraction) were determined by an Inductively Coupled Plasma Optical Emission Spectrometer (ICP-OES, ELAN DRC-e, Perkin Elmer, Waltham, MA, USA) due to their high concentrations in solutions. Cd concentration was measured by an Inductively Coupled Plasma Mass Spectrometry (ICP-MS, Optima 5300 DV, Perkin Elmer) due to its low concentration in the extracts.

### 2.4. Parameter Estimation

To assess the risk of HM pollution in greenhouse soils in the Shouguang District, methods recommended by the Environmental Quality Evaluation Standard for Farmland of Greenhouse Vegetable Production (HJ333-2006) [31], which is implemented by the State Environmental Protection Administration of China, as well as the geo-accumulation index ($I_{\mathrm{geo}}$) [32] were used.

#### 2.4.1. Environmental Quality Evaluation Standard for Farmland of Greenhouse Vegetables Production (HJ333-2006)

According to pollutant toxicological characteristics, the HJ333-2006 method divides the evaluation indices into two categories: strict and general control indicators. The strict control indicator is evaluated according to the single quality index (SI) while the general control indicator is evaluated by the comprehensive quality index (CI). Cadmium, As, Hg, Pb, and Cr belong to the strict control indicator while Cu, Zn, and Ni belong to the general control indicator. Table 1 lists the limits of these HMs defined in the HJ333-2006 method.

The evaluation parameters mentioned above are calculated using the following equations:(1)SI= single measured value / single standard value
(2)CI = [√((Average SI) ^ 2+(Max SI) ^ 2 )]/2

The single measured value represents the value of a certain HM in soil and the single standard value represents the corresponding HM limit in soil. The contamination categories, based on the HJ333-2006 method, are classified as follows: safety (SI or CI < 0.7), slight safety (0.7 < SI or CI < 1), and pollution (SI or CI > 1).

#### 2.4.2. Geo-accumulation Index

The geo-accumulation index ($I_{\mathrm{geo}}$), originally proposed by Müller et al. [32], is widely used to assess the potential risks of metals that derive from anthropogenic activities. The $I_{\mathrm{geo}}$ is calculated with the following equation: (3)$$I_{\mathrm{geo}}{=\log}_{2}{[C}_{n}/(1{.5\;\times \;B}_{n})]$$
where $C_{n}$ denotes the concentration of metal n in the soil and $B_{n}$ represents the corresponding background value. In this study, the metal concentrations in the cinnamon soils of Shouguang were used as the background values (Table 2). A constant of 1.5 was used to analyze natural fluctuations and detect minuscule anthropogenic influences. A positive $I_{\mathrm{geo}}$ indicates that there is a large HM contribution from anthropogenic activities. The contamination categories, based on the $I_{\mathrm{geo}}$, are further classified as follows: uncontaminated ($I_{\mathrm{geo}}$ < 0), uncontaminated to moderately contaminated (0 < $I_{\mathrm{geo}}$ < 1), moderately contaminated (1 < $I_{\mathrm{geo}}$ < 2), moderately to heavily contaminated (2 < $I_{\mathrm{geo}}$ < 3), heavily contaminated (3 < $I_{\mathrm{geo}}$ < 4), heavily to extremely contaminated (4 < $I_{\mathrm{geo}}$ < 5), and extremely contaminated ($I_{\mathrm{geo}}$ > 5).

### 2.5. Data Analysis

Data analysis was performed using the SPSS 23.0 statistical software (International Business Machines Corporation, Armonk, NY, USA). The sampling map and $I_{\mathrm{geo}}$ of each site were charted by ArcGIS 10.2 software (Environmental Systems Research Institute, Redlands, CA, USA). Pearson correlation analysis was used to determine relationships between different HMs, as well as relationships between HM concentrations and selected basic edaphic parameters, such as SOC, total phosphorus (P), total potassium (K), and aluminum and iron (Al_2_O_3_ and Fe_2_O_3_, i.e., the main constituents of geogenic and pedogenic materials in soils and measured by digesting soil samples), the data obtained are normally distributed and have 3 replications. To further identify associations among and common sources of metals, PCA was performed with a varimax rotation, which can facilitate interpreting the results by minimizing variable numbers with high loading on each component. Variables were not considered to identify source categories until their factor loadings (absolute magnitude) were >1.0.

The data that support the findings of this study are available from the corresponding author and first author on request.

## 3. Results and Discussion

### 3.1. HM Concentrations in Greenhouse Soils

The mean concentrations of Cd, Cr, Cu, Ni, Pb, and Zn in the greenhouse soils were 0.21, 70.7, 42.1, 23.0, 18.9 and 144.4 mg·kg^−1^, respectively (Table 2), which do not exceed the limits set by HJ333-2006 (Table 1). However, there are 5 samples exceed the Cd limit with an excessive rate of 16.1%, while other HMs in all samples do not exceed the limits. And compared with the corresponding natural background values in Shouguang (in mg·kg^−1^: Cd = 0.08; Cu = 23.6; Zn = 64.5) [10], the mean concentrations of Cd, Cu, and Zn significantly increased by 164.8%, 78.6%, and 123.9%, respectively. On the other hand, their coefficients of variation (CVs) were also relatively high, with a decreasing order of Cd (52.0%) > Cu (35.5%) > Zn (30.1%). Manta, et al. [33] suggested that anthropogenic activities lead to high HM concentrations and CVs, which suggests that Cd, Cu, and Zn accumulations in Shouguang are possibly due to agricultural activities. Meanwhile, the mean concentrations of Cr, Ni, and Pb in greenhouse soils were lower than or similar to their background values (in mg·kg^−1^: Cr = 66.0; Ni = 28.6; Pb = 26.0). Their low CVs (< 20%) further imply a limited influence from agricultural activities. Similarly, Cd, Cu, and Zn accumulations in greenhouse soils also exist in the Wuwei District and Yellow River Irrigation Region in northwest China [4,34]. However, compared with other regions in China, the mean Cd concentration in the Shouguang greenhouse soils was relatively low, only higher than those in the Nanjing and Wuqing Districts, while Cu and Zn concentrations were higher, except for the Zn in Nanjing and Yunnan (Table 3). In addition, compared with Spain and Turkey (due to rather few data sets of HM contents in greenhouse soils were obtained in other countries), the average Cd concentration, as well as the concentrations of Ni and Pb, in Shouguang were still relatively low while Cu and Zn exhibited opposite trends (Table 3). Overall, Cd, Cu, and Zn show general accumulations in greenhouse soils, with Shouguang exhibiting lower Cd accumulation and higher Cu and Zn accumulations.

### 3.2. Potential HM Contamination Risks in Greenhouse Soils

To further evaluate the HM pollution status, both the SI and CI indices (Table 4) were calculated. The SI values for six HMs in greenhouse soils from Shouguang were all less than 0.7, decreasing in the order of Cd (0.61) > Zn (0.53) > Cu (0.42) > Ni (0.42) > Pb (0.38). Based on the SI and CI (0.537) results, the contamination categories in Shouguang can be classified as safe and suitable for pollution-free vegetable production. Nonetheless, Cd exhibited a relatively higher environmental risk in the study area. As mentioned above, Cd had an over-standard rate of 16.1% and its maximum concentration was 43.7% higher than the limit (HJ333-2006), which is consistent with results from Zhang, et al. [44] and Wang, et al. [9] conducted in Baiyin (Gansu Province) and Shandong Province in China, respectively.

The $I_{\mathrm{geo}}$ results of each sampling site were showed in Figure 2, to clarify the HM accumulations and distributions in the greenhouse soils in Shouguang. In Figure 2, spatial distributions of $I_{\mathrm{geo}}$ for Cd, Cu, and Zn are distinctly different from Cr, Ni, and Pb, as well as the fact that Cd, Cu and Zn have relatively high spatial variations and high $I_{\mathrm{geo}}.$ Combined with the HM concentration distributions, Cd showed the highest contamination risk, with 30% of the sampling sites characterized as moderately contaminated ($I_{\mathrm{geo}}$ > 1), while Cu and Zn showed a slight contamination risk with over 50% of the sampling sites characterized as uncontaminated to moderately contaminated (0 < $I_{\mathrm{geo}}$ < 1). This is similar to the results reported in Tian et al. [12], who found that Cd accumulation was the most likely environmental problem in greenhouse soils of Shandong and Jiangsu provinces. Meanwhile, Cd, Cu, and Zn contaminations caused by anthropogenic sources has also been reported as a serious problem in greenhouse soils in southwestern China [13]. However, $I_{\mathrm{geo}}$ values were negative for Cr (except one site), Ni, and Pb at all sampling sites (Figure 2), indicating no Cr, Ni, or Pb pollutions.

### 3.3. Chemical Fractions of HMs in Greenhouse Soils

To further clarify environmental risks of HMs, the migration ability and bioavailability of HMs that is closely related to their chemical fractions in soils should be concern. Figure 3 shows the results of sequential chemical extraction, which are expressed as percentages of the sum of each chemical fraction.

In Shouguang, the dominant chemical fraction for Cd was the residual fraction (33.7% on average), followed by the exchangeable fraction (24.7% on average), carbonate-bound fraction (23.7% on average), Fe-Mn oxide-bound fraction (4.4% on average), and organically bound fraction (3.47% on average). Compared with other HMs, the proportion of Cd associated with the exchangeable fraction was the highest, ranging from 16.6–36.1%, which may have resulted from Cd’s strong polarization ability that led to its weak adsorption on the interface of soil organic matter [45]. Previous studies have also indicated that a high proportion of exchangeable Cd exists in most agricultural soils (including greenhouse soils), which would easily lead to higher environmental risks [46,47,48,49].

For Cu, the dominant chemical fraction was also the residual fraction (48.8% on average), followed by the organically bound fraction (16.5% on average), Fe–Mn oxide-bound fraction (14.9% on average), carbonate-bound fraction (10.9% on average), and exchangeable fraction (8.86% on average). Compared with other HMs, Cu is easily bound to oxidizable substances (including organic matter and sulfides) [50,51,52]. However, although Cu is a potentially less mobile metal, it can be released into surrounding environment with exposure to strong oxidative, reductive, or acidic soil conditions.

The majority of Zn was bound with the Fe-Mn oxide fraction (41.6% on average), followed by the residual fraction (33.8% on average), carbonate-bound fraction (17.8% on average), organically bound fraction (5.32% on average), and exchangeable fraction (1.48% on average). Zinc was easily bound to clay minerals, carbonates, or hydrous oxides [53,54,55,56,57], so that the highest proportion of Zn bound to Fe-Mn oxide was also found in Shouguang.

As the low-risk HMs, the majority of the Pb (78.7% on average) and Cr (86.9% on average) were found in the residual fraction, which is indicative of their strong combinations within mineral crystal structures and low migration risks in natural conditions [54]. In addition, Ni was mainly associated with the residual (32.5% on average) and Fe–Mn oxide-bound fractions (28.6% on average). Nickel had the larger proportion of residual fraction due to the ease with which it binds to aluminosilicate minerals [58]. Furthermore, Ni belongs to the same group on the periodic table as Fe and Mn, such that Ni can exist in the Fe-Mn oxide lattice via isomorphic substitution or adsorption and precipitation onto the Fe-Mn oxide surface [21].

The sum of four fractions, except residual fraction, for Cd, Cu, Zn, Pb, Cr, and Ni is 66.3%, 51.2%, 66.2%, 21.3%,13.1%, and 67.5%, respectively. These imply that the potential bioaccessibility increases in the order of Cr < Pb < Cu < Zn < Cd < Ni. Meanwhile, mobility factor (MF), sum of exchangeable and carbonate-bound fractions, show increasing order of Cr (0.00%) < Zn (19.2%) < Cu (19.8%) < Pb (20.4%) < Ni (23.4%) < Cd (48.4%). Although Ni has the highest potential bioavailability, it mainly exists in fractions like Fe-Mn oxide-bound and organically bound fractions. However, for Cd, it exhibits a high potential bioavailability and MF, which indicates a clearly high Cd bioavailability in Shouguang and its potential threat to human health and ecosystem safety.

### 3.4. Identification of the Main Sources Causing HM Accumulations

Since HMs in greenhouse soils could originate from different sources, such as sewage sludge, livestock manure, inorganic fertilizers, pesticides and other agricultural chemicals, irrigation water [22,59], identifying pollution sources is the key to reducing HM accumulations in greenhouse soils.

As shown in Figure 3, proportions of dominant Cd, Cu, and Zn fractions at different sampling sites varied widely. However, for Cr, Ni and Pb, the proportion for each fraction at the different sampling sites mostly showed little change. As previous studies have indicated, the variations in the HM fraction distributions may have mainly resulted from anthropogenic pollution influences. For example, Zong et al. [57] reported that the proportions of the HM fractions varied in a wide range, except for Cr with pedogenic origins. Guo et al. [46] also found that there was a clear spatial heterogeneity in the HM fraction distributions in the different regions of Changchun, which was caused by differences in the pollution sources and soil pH. Additionally, Acosta et al. [53] proposed that different pollution sources affect the HM fraction distributions in Spain. Furthermore, the differences in the pollution sources may result in the differences in proportions of a certain HM fraction [46,60]. Organically bound Cu and Fe–Mn oxide-bound Zn were the major chemical fractions of Cu and Zn in swine and cow manure, respectively [61,62], and higher amounts of exchangeable Cd were mostly found in soils with frequent applications of chemical fertilizers [63]. In Shouguang, it has problems with excessive application of fertilizers [28], mainly as livestock manures and N-P-K compound fertilizer (Appendix A). Thus, the varied proportions of organically bound Cu, Fe-Mn oxide-bound Zn, and exchangeable Cd should be related to agricultural activities, i.e., excessive fertilizer (livestock manures and chemical fertilizer) application. Additionally, for Cr, Ni and Pb, high residual fraction proportions and their little spatial heterogeneity in chemical fraction distribution should indicate their parent rock origins [57].

Furthermore, inter-element relationships can provide interesting information on metal element sources [64]. Pearson correlation coefficients were calculated between six HMs and related soil properties (Fe_2_O_3_, Al_2_O_3_, total K, total P, SOC, and cultivation age), and the results are shown in Table 5. A significant positive correlation (*p* < 0.05) was observed between Cd and Zn (r = 0.379) while significant positive correlations (*p* < 0.01) occur between Cu and Zn (r = 0.788), Cr and Ni (r = 0.686), Cr and Pb (r = 0.759), and Ni and Pb (r = 0.918). Significant positive correlations were also observed between Cd, Cu, and Zn with SOC and total P, with the following correlation coefficients: 0.419 (Cd-SOC, *p* < 0.05), 0.440 (Cu-SOC, *p* < 0.05), 0.578 (Zn-SOC, *p* < 0.01), 0.535 (Cd-P, *p* < 0.01), and 0.402 (Zn-P, *p* < 0.05). These findings are similar to the results of previous studies [1,65,66]. Furthermore, significant positive correlations (*p* < 0.01) between Cr, Ni, and Pb with Al_2_O_3_ and Fe_2_O_3_ were observed, with correlation coefficients of 0.783 and 0.732 for Cr with Al_2_O_3_ and Fe_2_O_3_, 0.697 and 0.641 for Ni with Al_2_O_3_ and Fe_2_O_3_, and 0.790 and 0.729 for Pb with Al_2_O_3_ and Fe_2_O_3_, respectively. Since positive correlations between different HMs imply similar origins [3], it is speculated that Cd, Cu, and Zn in greenhouse soil in Shouguang derive from an identical source while Cr, Ni, and Pb originate from different sources.

Since a single correlation analysis is not sufficient to determine the HM sources [67], PCA was also used. Appendix A lists PCA results and Figure 4 shows principal component loadings. In this study, due to the limitation of an eigenvalue >1, two principal components (PC1 and PC2) were obtained, which cumulatively explained 69.147% of the total variance in the greenhouse soils. After a varimax rotation, PC1 can explain 43.163% of the total variance, which is strongly and positively related to Cr (0.895), Ni (0.840), Pb (0.911), Al (0.925), Fe (0.884) and K (0.682). The PC2 component was characterized by high positive factor loadings for Cd (0.699), Cu (0.791), Zn (0.865) and P (0.670) and can explain 23.984% of the total variance. The high positive factor loading on an identical component strongly implies that these metals have a common source.

PC1 can be representative of geogenic and pedogenic contributions because mean concentrations of Cr, Ni, and Pb did not exceed or approach background values. Numerous previous studies [23,68,69] have shown that concentrations of Cr and Ni in soils are closely related to their contents in parent materials, which is consistent with PCA results of this study. In Shouguang, soil parent material was mostly alluvium, in which Ni and Cr concentrations were mainly affected by pedogenic processes [70]. The study area is far away from main roads and any factories or mining regions. The soils were also kept in a closed greenhouse environment, which prevented them from being contaminated by dry and wet atmospheric Pb deposition.

Thus, Pb in non-polluted soils should mainly be affected by parent materials [71]. Based on correlation analysis results, significant positive correlations of Cr, Ni, and Pb with Al_2_O_3_ and Fe_2_O_3_ (Table 5) further indicate natural origins for Cr, Ni, and Pb in greenhouse soils since Al_2_O_3_ and Fe_2_O_3_ are main constituents of geogenic and pedogenic materials. Thus, both PCA and correlation analysis results demonstrate natural origins for Cr, Ni, and Pb.

PC2 can be considered as anthropogenic origin, since mean concentrations of Cd, Cu, and Zn in greenhouse soils were respectively 165%, 78.6%, and 124% higher than their background values. In agricultural soils, these metals may originate from agricultural activities, such as irrigation, fertilization, and pesticide application [8,72]. According to field investigation results (Appendix A) and researches conducted by Zeng et al. [28], main chemical fertilizers applied in Shouguang are ammonium dihydrogen phosphate, urea, calcium superphosphate and N-P-K compound fertilizers, while organic fertilizers were mainly livestock manures, all leading to accumulations of SOC and P in soils. Significant positive correlations between Cd, Cu, and Zn with SOC and P (Table 5) further suggested that livestock manures and phosphorus fertilizers are main sources of Cd, Cu, and Zn in Shouguang. Moreover, significant positive correlations between Cd and P and the cultivation age (Table 5) further indicate that Cd accumulation in soils results from long-term phosphorus fertilizer application due to high Cd concentrations (up to 30 mg·kg^−1^) in phosphorus fertilizers [73,74,75]. In China, livestock manures is a significant HM contributor, which accounts for approximately 55%, 69%, and 51% of total Cd, Cu, and Zn accumulations in agricultural soils, respectively [59]. Bai et al. [41] also reported that Cd, Cu, and Zn accumulation in greenhouse soils in Siping, China are mainly due to application of organic and chemical fertilizers. An investigation of four cities in Shandong Province by Liu et al. [7] showed that fertilizers and agrochemicals are the main sources of Cd, Cu, and Zn accumulation in soils while the concentrations of Cr and Ni in the soils are mainly affected by the parent rocks. Similar results have also been obtained by Nicholson et al. [73], who reported that livestock manure and chemical fertilizers account for approximately 41%, 43%, and 42% of the total Cd, Cu, and Zn accumulated in agricultural soils in England and Wales. Thus, PC2 represents agricultural origin.

PCA results combined with correlation analysis results show that Cd, Cu, and Zn mainly derive from agricultural inputs, especially as livestock manures and chemical fertilizers which was also the main inputs in greenhouse soil in Shouguang (Appendix A). Thus, annual input of heavy metals through livestock manures and compound fertilizers to greenhouse soils in Shouguang was calculated (Appendix A). Suppose current Cd concentration and its increasing input rate were respectively 0.21 mg·kg^−1^ and 0.004 mg·kg^−1^·yr^−1^ in greenhouse soils in Shouguang, the time required to reach the maximum permeable Cd limit (i.e., 0.3 mg·kg^−1^ for soil from HJ333-2006) is only approximately 23 years, whereas, for Cu and Zn, the time required is approximately 51 and 42 years, indicating a relative higher environmental risk of Cd in greenhouse soils in Shouguang

## 4. Conclusions

In Shouguang District, Cd, Cu, and Zn, exhibited varying degrees of accumulations (i.e., increases by 165%, 78.6%, and 124%, respectively) compared with their background values. In addition, Cd and P concentrations increased with the cultivation age and 16.1% of the sampling sites exceeded the standard Cd limit defined in HJ333-2006. Despite these results, the overall environmental quality of the greenhouse soils in Shouguang is safe for production of non-polluted vegetables, based on the criteria defined in HJ333-2006.

Cadmium had the highest MF among six HMs and exhibited variable proportions of exchangeable fraction in Shouguang, indicating its high bioavailability and environmental risk. In addition, both the correlation analysis and PCA results suggest that agricultural activities, especially application of livestock manures and chemical fertilizers, result in Cd, Cu, and Zn accumulations in greenhouse soils, whereas parent rocks mainly determine Cr, Ni, and Pb concentrations.

In Shouguang, the time required to reach the maximum permeable Cd, Cu and Zn limits (HJ333-2006) in agricultural soils is approximately 23, 51 and 42 years respectively if keeping present annual Cd, Cu and Zn input rates from livestock manures and compound fertilizers. To reduce their accumulations in Shouguang greenhouse soils, significant attention should be paid to perform effective pollution management policies, such as formulation and implementation of stricter measures on fertilizer production and application rate. This study provide evidence that current farming practices have significant contributions to heavy metal accumulations in soils in Shouguang district.

## Figures and Tables

**Figure 1 ijerph-16-02805-f001:**
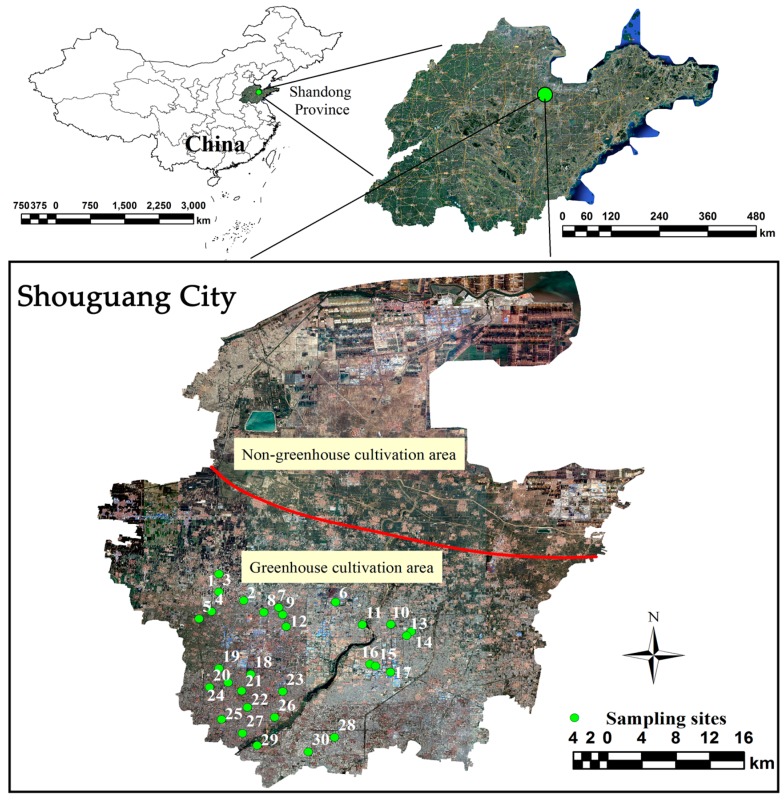
The distribution of sampling sites in the study region (denoted by green circles).

**Figure 2 ijerph-16-02805-f002:**
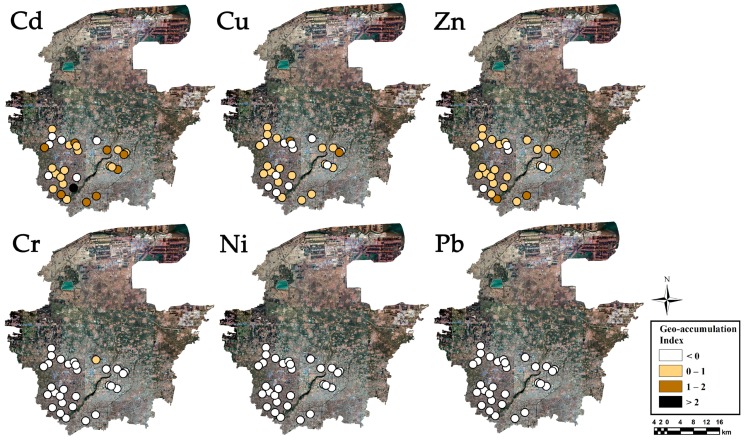
Spatial variations of $I_{\mathrm{geo}}$ index for HMs in greenhouse soils in Shouguang.

**Figure 3 ijerph-16-02805-f003:**
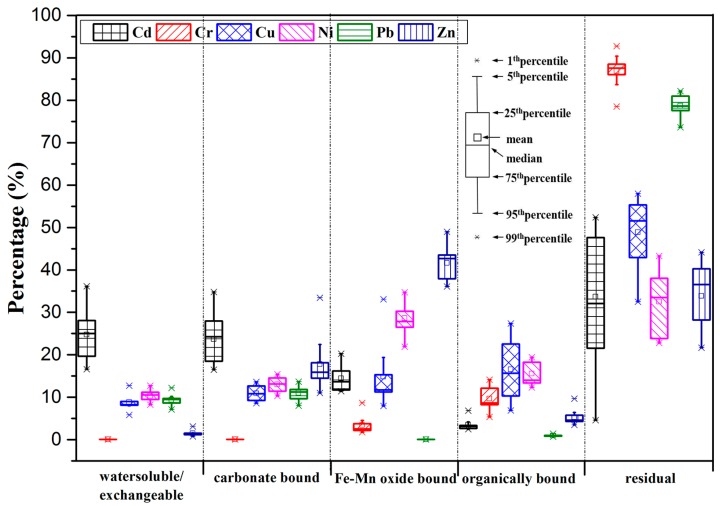
Chemical fractions of HMs (Cd, Cr, Cu, Ni, Pb, and Zn) in greenhouse soils in Shouguang.

**Figure 4 ijerph-16-02805-f004:**
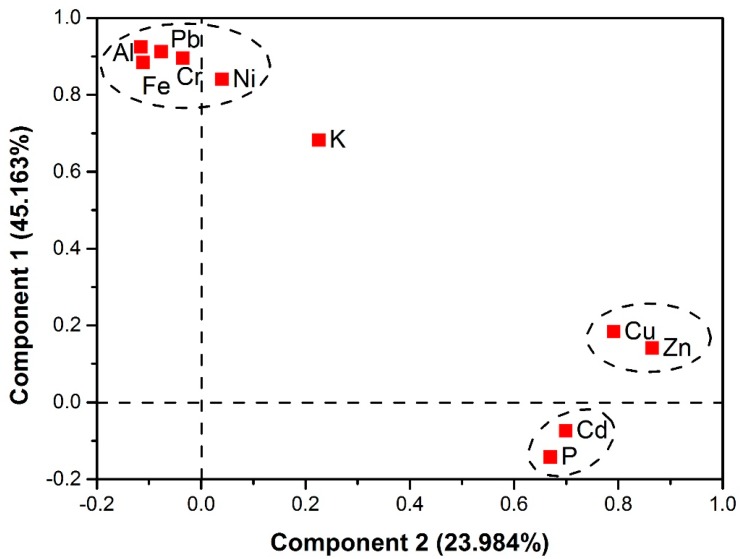
Factor loadings for two principal components after varimax rotation.

**Table 1 ijerph-16-02805-t001:** The HM limits based on the Environmental Quality Evaluation Standard for Farmland of Greenhouse Vegetables Production.

Soil pH	Cd	Cr	Cu	Ni	Pb	Zn
	mg·kg^−1^					
Soil pH < 6.5	0.30	150	50	40	50	200
6.5 < Soil pH < 7.5	0.30	200	100	50	50	250
Soil pH > 7.5	0.40	250	100	60	50	300

**Table 2 ijerph-16-02805-t002:** Descriptive statistics for soil properties and HM contents in the study area.

Parameter	Range (mg·kg^−1^)	Mean (mg·kg^−1^)	Median (mg·kg^−1^)	SD	CV	Skewness	Kurtosis	Background Value ^a^ (mg·kg^−1^)
Cd	0.09–0.48	0.21	0.17	0.11	52.0	1.12	0.39	0.08
Cr	37.0–109.1	70.7	68.4	16.1	22.7	0.31	−0.17	66.0
Cu	20.8–76.9	42.1	39.8	14.9	35.5	0.73	0.11	23.6
Ni	16.2–31.2	23.0	21.9	3.75	16.3	0.46	−0.62	28.6
Pb	14.5–25.7	18.9	18.2	2.89	15.3	0.67	−0.12	26.0
Zn	75.9–246.1	144.4	139.0	43.5	30.1	0.76	0.56	64.5
pH	6.6–7.86	7.39	7.52	0.38	5.1	−0.95	−0.33	-
SOC (%)	1.4–3.0	2.0	2.1	3.71	18.2	0.09	−0.23	-

*SOC* Soil organic carbon (%), *SD* standard deviation, *CV* coefficient of variation (in %). ^a^ Background values for soils in Shouguang from Liu et al. [10].

**Table 3 ijerph-16-02805-t003:** Comparisons of HM concentrations in greenhouse soils from various regions.

Study Area		Cd	Cr	Cu	Ni	Pb	Zn	Reference
		mg·kg^−1^						
Spain	Western Almería	1.20	- ^a^	- ^a^	38.6	69.9	- ^a^	[3]
	Granada and Almería	1.10	50.3	30.2	36.0	68.9	133	[6]
Turkey	Antalya Aksu	- ^a^	- ^a^	25.0	16.6	45.0	88.0	[35]
	Çanakkale	1.07	85.5	46.7	55.5	19.6	84.6	[36]
China	North China Plain	0.64	87.9	29.9	- ^a^	29.4	82.4	[37]
	Nanjing, Jangsu	0.15	- ^a^	52.2	- ^a^	53	117	[38]
	Yunnan	0.40	89.9	46.6	18.3	19.8	48.6	[39]
	Lanzhou, Gansu	0.26	65.5	-^a^	12.3	23.9	- ^a^	[40]
	Siping, Jilin	0.47	67.5	37	25.2	18.0	87.7	[41]
	Wuwei, Gansu	0.42	67.9	33.9	28.6	20.8	85.3	[4]
	Wuqing, Tianjin	0.14	62.5	24.5	32.1	24.3	71.6	[42]
	Henan	- ^a^	54.4	32.0	37.7	- ^a^	49.0	[43]
	Shouguang, Shandong	0.21	70.7	42.1	23.0	18.9	144.4	This study

^a^ Not available.

**Table 4 ijerph-16-02805-t004:** The single and comprehensive quality indices for HMs in greenhouse soils in Shouguang.

Element		Cd	Cr	Cu	Ni	Pb	Zn
Single Quality Index	Mean	0.615	0.315	0.421	0.419	0.378	0.526
	SD	0.335	0.087	0.149	0.095	0.058	0.184
	Range	0.224–1.50	0.148–0.475	0.208–0.769	0.269–0.624	0.291–0.513	0.253–0.984
	Over-standardrate (%)	16.13	0	0	0	0	0
Comprehensive Quality Index		0.537					

**Table 5 ijerph-16-02805-t005:** The Pearson Correlation coefficients (r) between the HM contents and soil property parameters in greenhouse soils in Shouguang.

Item	Cd	Cr	Cu	Ni	Pb	Zn	Fe_2_O_3_	Al_2_O_3_	K	P	SOC	Cultivation Age
Cd	1											
Cr	−0.048	1										
Cu	0.336	0.139	1									
Ni	−0.022	0.686 **	0.134	1								
Pb	−0.097	0.759 **	0.075	0.918 **	1							
Zn	0.379 *	0.013	0.788 **	0.221	0.094	1						
Fe_2_O_3_	−0.043	0.732 **	0.038	0.641 **	0.729 **	−0.028	1					
Al_2_O_3_	−0.118	0.783 **	0.030	0.697 **	0.790 **	0.028	0.923 **	1				
K	−0.005	0.673 **	0.323	0.330	0.468 **	0.223	0.544 **	0.575 **	1			
P	0.535 **	−0.129	0.194	−0.080	−0.183	0.402 *	−0.136	−0.141	0.065	1		
SOC	0.419 *	0.224	0.440 *	0.231	0.142	0.578 **	0.090	0.144	0.379 *	0.566 **	1	
Cultivation age	0.647 **	−0.111	−0.097	−0.200	−0.214	−0.025	0.001	−0.091	−0.132	0.560 **	0.072	1

* Correlations are significant at the 0.05 level. ** Correlations are significant at the 0.01 level.

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
