# Peer review of "Cd, Cu, and Zn Accumulations Caused by Long-Term Fertilization in Greenhouse Soils and Their Potential Risk Assessment"

_ijerph, 2019, doi:10.3390/ijerph16152805_

Round 1

Reviewer 1 Report

The manuscript entitled “The Effects of Long-term Fertilization on Cd, Cu, and Zn Accumulation in Greenhouse Soils from the Shouguang District, Shandong Province, China” by Liao et al. presents an study on the concentration, distribution and sources of heavy metals in greenhouse soils in Shouguang (China), estimating the time required to reach the maximum limits considered safe for production of vegetables. Although it is already known that long-term intensive fertilization using chemical fertilizers and livestock manure leads to heavy metal accumulation, mainly Cd (by phosphate fertilizers) and Cu, and Zn (and As) (by livestock manure), this is a right and well-structured work that supplies related specific information of the area subject of study. In any case, some minor clarifications and corrections are necessary before accepting this manuscript for publication. Revision should be focused on the following points:

On page 4, in line 129, in relation to equation (1), please, specify which are the single standard values or clearly indicate where such values are reflected.

On page 6, in line 185, please delete the word “results” appearing after the word “extraction”.

On page 6, in lines 219-220, it is stated that the exchangeable fraction represents weakly bound metals that are soluble in water. The whole exchangeable fraction is not soluble in water, only the soluble fraction is released or extracted by water. Please, revise and correct.

In Table 7, please, write the number “3” appearing in the units of the Input intensity in the upper script form.

In Table 7, the expression of units employs the symbol “/”, whereas in the text the expression of units employs the symbol “-1”. Please, unify.

Author Response

We much appreciate the review comments, which will substantially improve the quality of our manuscript. Please see the attachment.

Reviewer 2 Report

Very good effort. Some comments are given to the writers:

GENERAL COMMENT

The research would be more complete if there were measurements of heavy metals in growing vegetables in the area (to see the impact on humans).

Other comments

-Tables 2,3,6 should be redesigned

-  Bibliography needs rewriting using the same format

- Bibliography in the text missing at the end e.g. Tian et al, 2018 (line 48), Sacchi and Mallen 2001 (line 67), Muller 1979 (line 133), Zeng et al 2016 (line 247), Xu et al 2017 (line 249)

- Bibliography at the end is missing from the text e.g. 19, 24, 45, 52, 70

- Bibliography 25 and 27 in the text is the same Liu et al should be written in other way e.g. a, b

- Bibliography line 218 in the text Jing et al 2010 at the end Jin

- line 131 pollution (SI or CI ˂1) →(SI or CI ˃1)  ?

Author Response

(The authors gave the same response as above.)

Reviewer 3 Report

Dear Authors,

The paper has a good initiative in terms of assessing the heavy metals contamination and pollution in the greenhouse soil. However, the flow of the story is not compelling and need some improvement. Also, the design of the experiment and the data analysis section need to be improved with some background information on the topic. Please see the details comments made on the PDF attached.

Thanks 

Author Response

(The authors gave the same response as above.)

Reviewer 4 Report

The research covers a typical soil pollution characterization by heavy metals in the greenhouses of Shouguang district in China. The research has certain lacks but it is generally well structured as typical reports of this nature, and findings are explained according to modern literature.

However, from the scientific point of view, it does not present any novelty beyond the possible interest that local readers may have on knowing the quality of soils in greenhouses of this district. So I recommend its publication only if the journal is interested in publishing case studies.

General comments:

1-      Firstly, avoid the use of personal expression like “we” “us” “our”, etc. There are a lot in the text.

2-      Table headings are broken. (e.g. skewne - ss, kurtosi – s, tota – l). Words should be on the same line and this happens in more than one table during all the text. If it is not problem of the compression of the pdf, correct it please.

Specific comments

Figure 1 – A minimap showing the position of Shouguang within China would be optimal.

Line 92 – What was the criterion to choose these thirty sites? I guess it was random but it is not specified.

Table 3 should include the times the legal limits are exceeded per element. Regarding the explanation, I see that values for Cd, Cu and Zn surpass sometimes the Chinese legal limits (table 1), the rest of HM don’t. Is the problem a general concern or just specific of extreme values? For both cases, it should be explained later on the text to ascertain if this is specific of an area, a punctual greenhouse pollution, or if the pollution is generalized

Line 173 – Here you compare greenhouse soils in China with other greenhouses of Turkey and Spain. Why these? Are these data comparable? Is the harvest or the fertiliziers used in the areas the same? Be careful when comparing areas that are so different and distanced because they are hardly comparable.

Section 3.2 – There is so much text in this and all this can be summarized in a table and results explained on the text for a better interpretation. To improve the understanding, I suggest focusing first on elements of highest concern (Cd, Cu, Zn) and then on lower ones.

With the Igeo you are providing maps of interpolation in an area. This is not correct since soils analyzed belong to greenhouses. Thus, each point is a greenhouse, a isolated area where any farmer may introduce his/her fertilizers or specific treatment to the soil. Out of the greenhouses soil is exposed to other factors. So HM content out of them are not predictabe. Thus, from my point of view it would be better to include just the punctual value of the Igeo rather than interpolating. Moreover, in case you decide to interpolate, samples seem too far one from the other (km) to perform a kriging interpolation. And the variogram has a maximum reach of operability. This is not expressed in the materials and methods section. Have you computed the variogram? What parameters did you introduce? Please introduce the variogram on the text. Which software did you use to compute the maps? ArcGIS, isn’t it? This section may be highly improved.

A PCA carried out with just six elements is a bit weak to perform conclusions. Why not including other data from the bivariate analysis? Line 352 is not well explained- A PCA alone cannot explain that a HM derives from anthropogenic issues, you conclude it from other facts. Review this section, please.

Author Response

(The authors gave the same response as above.)

Round 2

Reviewer 3 Report

Dear Authors,

Thank you very much for considering all the comments in the revised version of this manuscript. This paper is now suitable for publication. Please look for two more observations to be addressed in lines 108, 132, 190, and 193.

Regards

Author Response

Thank you for your good suggestions. Please see the attachment.

Reviewer 4 Report

The authors have introduced multiple changes in the article and also modified the suggestions I made so in my opinion the article is ready to publication

Author Response

Thank you for your review comments, which substantially improve the quality of our manuscript.